# Lie-Access Neural Turing Machines

**Greg Yang and Alexander M. Rush**
{gyang@college,srush@seas}.harvard.edu
Harvard University
Cambridge, MA 02138, USA

## Abstract

External neural memory structures have recently become a popular tool for algorithmic deep learning (Graves et al., 2014; Weston et al., 2014). These models generally utilize differentiable versions of traditional discrete memory-access structures (random access, stacks, tapes) to provide the storage necessary for computational tasks. In this work, we argue that these neural memory systems lack specific structure important for relative indexing, and propose an alternative model, Lie-access memory, that is explicitly designed for the neural setting. In this paradigm, memory is accessed using a continuous head in a key-space manifold. The head is moved via Lie group actions, such as shifts or rotations, generated by a controller, and memory access is performed by linear smoothing in key space. We argue that Lie groups provide a natural generalization of discrete memory structures, such as Turing machines, as they provide inverse and identity operators while maintaining differentiability. To experiment with this approach, we implement a simplified Lie-access neural Turing machine (LANTM) with different Lie groups. We find that this approach is able to perform well on a range of algorithmic tasks.

## 1 Introduction

Recent work on neural Turing machines (NTMs) (Graves et al., 2014; 2016) and memory networks (MemNNs) (Weston et al., 2014) has repopularized the use of explicit external memory in neural networks and demonstrated that these networks can be effectively trained in an end-to-end fashion. These methods have been successfully applied to question answering (Weston et al., 2014; Sukhbaatar et al., 2015; Kumar et al., 2015), algorithm learning (Graves et al., 2014; Kalchbrenner et al., 2015; Kaiser & Sutskever, 2015; Kurach et al., 2015; Zaremba & Sutskever, 2015; Grefenstette et al., 2015; Joulin & Mikolov, 2015), machine translation (Kalchbrenner et al., 2015), and other tasks. This methodology has the potential to extend deep networks in a general-purpose way beyond the limitations of fixed-length encodings such as standard recurrent neural networks (RNNs).

A shared theme in many of these works (and earlier exploration of neural memory) is to re-frame traditional memory access paradigms to be continuous and possibly differentiable to allow for back-propagation. In MemNNs, traditional *random-access* memory is replaced with a ranking approach that finds the most likely memory. In the work of Grefenstette et al. (2015), classical *stack-, queue-*, and *deque-based* memories are replaced by soft-differentiable stack, queue, and deque data-structures. In NTMs, *sequential* local-access memory is simulated by an explicit tape data structure.

This work questions the assumption that neural memory should mimic the structure of traditional discrete memory. We argue that a neural memory should provide the following: (A) differentiability for end-to-end training and (B) robust *relative* indexing (perhaps in addition to random-access). Surprisingly many neural memory systems fail one of these conditions, either lacking Criterion B, discussed below, or employing extensions like REINFORCE to work around lack of differentiability (Zaremba & Sutskever, 2015).

We propose instead a class of memory access techniques based around Lie groups, i.e. groups with differentiable operations, which provide a natural structure for neural memory access. By definition, their differentiability satisfies the concerns of Criterion A. Additionally the group axioms provide identity, invertibility, and associativity, all of which are desirable properties for a relative indexing scheme (Criterion B), and all of which are satisfied by standard Turing machines. Notably though,

simple group properties like invertibility are not satisfied by *neural* Turing machines, differentiable neural computers, or even by simple soft-tape machines. In short, in our method, we construct memory systems with keys placed on a manifold, and where relative access operations are provided by Lie groups.

To experiment with this approach, we implement a neural Turing machine with an LSTM controller and several versions of Lie-access memory, which we call Lie-access neural Turing machines (LANTM). The details of these models are exhibited in Section 4.[1] Our main experimental results are presented in Section 5. The LANTM model is able to learn non-trivial algorithmic tasks such as copying and permutating sequences with higher accuracy than more traditional memory-based approaches, and significantly better than fixed memory LSTM models. The memory structures and key transformation learned by the model resemble interesting continuous space representations of traditional discrete memory data structures.

## 2 BACKGROUND: RECURRENT NEURAL NETWORKS WITH MEMORY

This work focuses particularly on recurrent neural network (RNN) controllers of abstract neural memories. Formally, an RNN is a differentiable function RNN : $\mathcal{X} \times \mathcal{H} \to \mathcal{H}$, where $\mathcal{X}$ is an arbitrary input space and $\mathcal{H}$ is the hidden state space. On input $(x^{(1)}, \ldots, x^{(T)}) \in \mathcal{X}^T$ and with initial state $h^{(0)} \in \mathcal{H}$, the RNN produces states $h^{(1)}, \ldots, h^{(T)}$ based on the recurrence,

$$h^{(t)} := \text{RNN}(x^{(t)}, h^{(t-1)}).$$

These states can be used for downstream tasks, for example sequence prediction which produces outputs $(y^{(1)}, \ldots, y^{(T)})$ based on an additional transformation and prediction layer $y^{(t)} = F(h^{(t)})$ such as a linear-layer followed by a softmax. RNNs can be trained end-to-end by backpropagation-through-time (BPTT) (Werbos, 1990). In practice, we use long short-term memory (LSTM) RNNs (Hochreiter & Schmidhuber, 1997). LSTM's hidden state consists of two variables $(c^{(t)}, h^{(t)})$, where $h^{(t)}$ is also the output to the external world; we however use the above notation for simplicity.

An RNN can also serve as the controller for an external memory system (Graves et al., 2014; Grefenstette et al., 2015; Zaremba & Sutskever, 2015), which enables: (1) the entire system to carry state over time from both the RNN and the external memory, and (2) the RNN controller to collect readings from and compute additional instructions to the external memory. Formally, we extend the recurrence to,

$$h^{(t)} := \text{RNN}([x^{(t)}; \rho^{(t-1)}], h^{(t-1)}),$$
$$\Sigma^{(t)}, \rho^{(t)} := \text{RW}(\Sigma^{(t-1)}, h^{(t)}),$$

where $\Sigma$ is the abstract memory state, and $\rho^{(t)}$ is the value read from memory, and $h$ is used as an abstract controller command to a read/write function RW. Writing occurs in the mutation of $\Sigma$ at each time step. Throughout this work, $\Sigma$ will take the form of an ordered set $\{(k_i, v_i, s_i)\}_i$ where $k_i \in \mathcal{K}$ is an arbitrary key, $v_i \in \mathbb{R}^m$ is a memory value, and $s_i \in \mathbb{R}^+$ is a memory strength.

In order for the model to be trainable with backpropagation, the memory function RW must also be differentiable. Several forms of differentiable memory have been proposed in the literature. We begin by describing two simple forms: (neural) random-access memory and (neural) tape-based memory. For this section, we focus on the read step and assume $\Sigma$ is fixed.

**Random-Access Memory**  Random-access memory consists of using a now standard attention-mechanism or MemNN to read a memory (our description follows Miller et al. (2016)). The controller hidden state is used to output a random-access pointer, $q'(h)$ that determines a weighting of memory vectors via dot products with the corresponding keys. This weighting in turn determines the read values via linear smoothing based on a function $w$,

$$w_i(q, \Sigma) := \frac{s_i \exp \langle q, k_i \rangle}{\sum_j s_j \exp \langle q, k_j \rangle} \qquad\qquad \rho := \sum_i w_i(q'(h), \Sigma) v_i.$$

The final read memory is based on how "close" the read pointer was to each of the keys, where closeness in key space is determined by $w$.

---

[1]Our implementations are available at `https://github.com/harvardnlp/lie-access-memory`

**Tape-Based Memory** Neural memories can also be extended to support relative access by maintaining read state. Following notation from Turing machines, we call this state the *head*, $q$. In the simplest case the recurrence now has the form,

$$\Sigma', q', \rho = \text{RW}(\Sigma, q, h),$$

and this can be extended to support multiple heads.

In the simplest case of soft tape-based memory (a naive version of the much more complicated neural Turing machine), the keys $k_i$ indicate one-hot positions along a tape with $k_i = \delta_i$. The head $q$ is a probability distribution over tape positions. It determines the read value by directly specifying the weights. The controller can only "shift" the head by outputting a kernel $K(h) = (K_{-1}, K_0, K_{+1})$ in the probability simplex $\Delta^2$ and applying convolution.

$$q'(q, h) := q * K(h), \qquad \text{i.e.} \qquad q'_j = q_{j-1}K_{+1} + q_j K_0 + q_{j+1} K_{-1}$$

We can view this as the soft version of a single-step discrete Turing machine where the kernel can softly shift the "head" of the machine one to the left, one to the right, or remain in the same location. The value returned can then be computed with linear smoothing as above,

$$w_i(q, \Sigma) := \frac{s_i \langle q, k_i \rangle}{\sum_j s_j \langle q, k_j \rangle} \qquad\qquad \rho := \sum_i w_i(q'(q, h), \Sigma) v_i.$$

## 3  LIE GROUPS FOR MEMORY

Let us now take a brief digression and consider the standard (non-neural) Turing machine (TM) and the movement of its head over a tape. A TM has a head $q \in \mathbb{Z}$ indicating the position on a tape. Between reads, the head can move any number of steps left or right. Moving $a + b$ steps and then $c$ steps eventually puts the head at the same location as moving $a$ steps and then $b + c$ steps — i.e. the head movement is *associative*. In addition, the machine should be able to reverse a head shift, for example, in a stack simulation algorithm, going from push to pop — i.e. each head movement should also have a corresponding *inverse*. Finally, the head should also be allowed to stay put, for example, to read a single data item and use it for multiple time points, an *identity*.

These movements correspond directly to group actions: the possible head movements should be associative, and contain inverse and identity elements. This group acts on the set of possible head locations. In a TM, the set of $\mathbb{Z}$-valued head movement acts on the set of locations on the $\mathbb{Z}$-indexed infinite tape. By our reasoning above, if a Turing machine is to store data contents at points in a general space $\mathcal{K}$ (instead of an infinite $\mathbb{Z}$-indexed tape), then its head movements should form a group and act on $\mathcal{K}$ via group actions.

For a neural memory system, we desire the network to be (almost everywhere) differentiable. The notion of "differentiable" groups is well-studied in mathematics, where they are known as *Lie groups*, and "differentiable group actions" are correspondingly called *Lie group actions*. In our case, using Lie group actions as generalized head movements on a general key space (more accurately, manifolds) would most importantly mean that we can take derivatives of these movements and perform the usual backpropagation algorithm.

## 4  LIE-ACCESS NEURAL TURING MACHINES

These properties motivate us to propose Lie access as an alternative formalism to popular neural memory systems, such as probabilistic tapes, which surprisingly do not satisfy invertibility and often do not provide an identity.[2] Our Lie-access memory will consist of a set of points in a manifold $\mathcal{K}$.

---

[2]The Markov kernel convolutional soft head shift mechanism proposed in Graves et al. (2014) and sketched in Section 2 does not in general have inverses. Indeed, the authors reported problems with the soft head losing "sharpness" over time, which they dealt with by sharpening coefficients. In the followup work, Graves et al. (2016) utilize a *temporal memory link matrix* for actions. They note, "the operation $Lw$ smoothly shifts the focus forwards to the locations written ... whereas $L^\top w$ shifts the focus backwards" but do not enforce this as a true inverse. They also explicitly do not include an identity, noting "Self-links are excluded (the diagonal of the link matrix is always 0)"; however, they could ignore the link matrix with an interpolation gate, which in effect acts as the identity.

We replace the discrete head with a continuous head $q \in \mathcal{K}$. The head moves based on a set of Lie group actions $a \in \mathcal{A}$ generated by the controller. To read memories, we will rely on a distance measure in this space, $d : \mathcal{K} \times \mathcal{K} \to \mathbb{R}^{\geq 0}$.[3] Together these properties describe a general class of possible neural memory architectures.

Formally a Lie-access neural Turing machine (LANTM) computes the following function,

$$\Sigma', q', q'_{(w)}, \rho := \mathrm{RW}(\Sigma, q, q_{(w)}, h)$$

where $q, q_{(w)} \in \mathcal{K}$ are resp. read and write heads, and $\Sigma$ is the memory itself. We implement $\Sigma$, as above, as a weighted dictionary $\Sigma = \{(k_i, v_i, s_i)\}_i$.

## 4.1 ADDRESSING PROCEDURE

The LANTM maintains a read head $q$ which at every step is first updated to $q'$ and then used to read from the memory table. This update occurs by selecting a Lie group action from $\mathcal{A}$ which then acts smoothly on the key space $\mathcal{K}$. We parametrize the action transformation, $a : \mathcal{H} \mapsto \mathcal{A}$ by the hidden state to produce the Lie action, $a(h) \in \mathcal{A}$. In the simplest case, the head is then updated based on this action (here $\cdot$ denotes group action): $q' := a(h) \cdot q$.

For instance, consider two possible Lie groups:

(1) A shift group $\mathbb{R}^2$ acting additively on $\mathbb{R}^2$. This means that $\mathcal{A} = \mathbb{R}^2$ so that $a(h) = (\alpha, \beta)$ acts upon a head $q = (x, y)$ by,

$$a(h) \cdot q = (\alpha, \beta) + (x, y) = (x + \alpha, y + \beta).$$

(2) A rotation group $SO(3)$ acting on the sphere $S^2 = \{v \in \mathbb{R}^3 : \|v\| = 1\}$. Each rotation can be described by its axis $\xi$ (a unit vector) and angle $\theta$. An action $(\xi, \theta) \cdot q$ is just the appropriate rotation of the point $q$, and is given by Rodrigues' rotation formula,

$$a(h) \cdot q = (\xi, \theta) \cdot q = q \cos \theta + (\xi \times q) \sin \theta + \xi \langle \xi, q \rangle (1 - \cos \theta).$$

Here $\times$ denotes cross product.

## 4.2 READING AND WRITING MEMORIES

Recall that memories are stored in $\Sigma$, each with a key, $k_i$, memory vector, $v_i$, and strength, $s_i$, and that memories are read using linear smoothing over vectors based on a key weighting function $w$, $\rho := \sum_i w_i(q', \Sigma) v_i$. While there are many possible weighting schemes, we use one based on the distance of each memory address from the head in key-space assuming a metric $d$ on $\mathcal{K}$. We consider two different weighting functions (1) inverse-square and (2) softmax. There first uses the polynomial law and the second an annealed softmax of the squared distances:

$$w_i^{(1)}(q, \Sigma) := \frac{s_i d(q, k_i)^{-2}}{\sum_j s_j d(q, k_j)^{-2}} \qquad w_i^{(2)}(q, \Sigma, T) := \frac{s_i \exp(-d(q, k_i)^2 / T)}{\sum_j s_j \exp(-d(q, k_j)^2 / T)},$$

where we use the convention that it takes the limit value when $q \to k_i$ and $T$ is a *temperature* that represents the certainty of its reading, i.e. higher $T$ creates more uniform $w$.

The writing procedure is similar to reading. The LANTM maintains a separate *write head* $q_{(w)}$ that moves analogously to the read head, i.e. with action function $a_{(w)}(h)$ and updated value $q'_{(w)}$. At each call to RW, a new memory is automatically appended to $\Sigma$ with $k = q'_{(w)}$. The corresponding

---

[3]This metric should satisfy a compatibility relation with the Lie group action. When points $x, y \in X$ are simultaneously moved by the same Lie group action $v$, their distance should stay the same (One possible mathematical formalization is that $X$ should be a Riemannian manifold and the Lie group should be a subgroup of $X$'s isometry group.): $d(vx, vy) = d(x, y)$. This condition ensures that if the machine writes a sequence of data along a "straight line" at points $x, vx, v^2 x, \ldots, v^k x$, then it can read the same sequence by emitting a read location $y$ close to $x$ and then follow the "$v$-trail" $y, vy, v^2 y, \ldots, v^k y$.

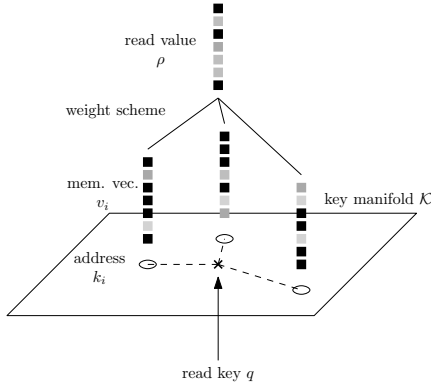

Figure 1: Retrieval of value from memory via a key. Weightings with unit sum are assigned to different memories depending on the distances from the addresses to the read key. Linear smoothing over values is used to emit the final read value. Both inverse-square and softmax schemes follow this method, but differ in their computations of the weightings.

memory $v$ and strength $s$ are created by MLP's $v(h) \in \mathbb{R}^m$ and $s(h) \in [0, 1]$ taking $h$ as input. After writing, the new memory set is,

$$\Sigma' := \Sigma \cup \{(q'_{(w)}, v(h), s(h))\}.$$

No explicit erase mechanism is provided, but to erase a memory $(k, v, s)$, the controller may in theory write $(k, -v, s)$.

### 4.3 COMBINING WITH RANDOM ACCESS

Finally we combine this relative addressing procedure with direct random-access to give the model the ability for absolute address access. We do this by outputting an absolute address each step and simply interpolating with our current head. Write $t(h) \in [0, 1]$ for the interpolation gate and $\tilde{q}(h) \in \mathcal{K}$ for our proposed random-access layer. For key space manifolds $\mathcal{K}$ like $\mathbb{R}^n$, [4] there's a well defined straight-line interpolation between two points, so we can set

$$q' := a \cdot (tq + (1 - t)\tilde{q})$$

where we have omitted the implied dependence on $h$. For other manifolds like the spheres $S^n$ that have well-behaved projection functions $\pi : \mathbb{R}^n \to S^n$, we can just project the straight-line interpolation to the sphere:

$$q' := a \cdot \pi(tq + (1 - t)\tilde{q}).$$

In the case of a sphere $S^n$, $\pi$ is just $L_2$-normalization.[5]

## 5 EXPERIMENTS

We experiment with Lie-access memory on a variety of algorithmic learning tasks. We are particularly interested in: (a) how Lie-access memory can be trained, (b) whether it can be effectively utilized for algorithmic learning, and (c) what internal structures the model learns compared to systems based directly on soft discrete memory. In particular Lie access is not equipped with an explicit stack or tape, so it would need to learn continuous patterns that capture these properties.

**Setup.** Our experiments utilize an LSTM controller in a version of the encoder-decoder setup (Sutskever et al., 2014), i.e. an encoding input pass followed by a decoding output pass. The encoder reads and writes memories at each step; the decoder only reads memories. The encoder is given $\langle s \rangle$,

---

[4] Or in general, manifolds with convex embeddings in $\mathbb{R}^n$.
[5] Technically, in the sphere case, $\text{dom } \pi = \mathbb{R}^d - \{0\}$. But in practice one almost never gets 0 from a straight-line interpolation, so computationally this makes little difference.

followed by an the input sequence, and then $\langle /s \rangle$ to terminate input. The decoder is not re-fed its output or the correct symbol, i.e. we do not use teacher forcing, so $x^{(t)}$ is a fixed placeholder input symbol. The decoder must correctly emit an end-of-output symbol $\langle /e \rangle$ to terminate.

**Models and Baselines.** We implement three main baseline models including: (a) a standard *LSTM* encoder-decoder, without explicit external memory, (b) a random access memory network, *RAM* using the key-value formulation as described in the background, roughly analogous to an attention-based encoder-decoder, and (c) an interpolation of a *RAM/Tape*-based memory network as described in the background, i.e. a highly simplified version of a true NTM (Graves et al., 2014) with a sharpening parameter. Our models include four versions of Lie-access memory. The main model, *LANTM*, has an LSTM controller, with a shift group $\mathcal{A} = \mathbb{R}^2$ acting additively on key space $\mathcal{K} = \mathbb{R}^2$. We also consider a model *SLANTM* with spherical memory, utilizing a rotation group $\mathcal{A} = SO(3)$ acting on keys in the sphere $\mathcal{K} = S^2$. For both of the models, the distance function $d$ is the Euclidean ($L_2$) distance, and we experiment with smoothing using *inverse-square* (default) and with an annealed *softmax*.[6]

**Model Setup.** For all tasks, the LSTM baseline has 1 to 4 layers, each with 256 cells. Each of the other models has a single-layer, 50-cell LSTM controller, with memory width (i.e. the size of each memory vector) 20. Other parameters such as learning rate, decay, and intialization are found through grid search. Further hyperparameter details are give in the appendix.

**Tasks.** Our experiments are on a series of algorithmic tasks shown in Table 1a. The COPY, RE-VERSE, and BIGRAM FLIP tasks are based on Grefenstette et al. (2015); the DOUBLE and INTER-LEAVED ADD tasks are designed in a similar vein. Additionally we also include three harder tasks: ODD FIRST, REPEAT COPY, and PRIORITY SORT. In ODD FIRST, the model must output the odd-indexed elements first, followed by the even-indexed elements. In REPEAT COPY, each model must repeat a sequence of length 20, $N$ times. In PRIORITY SORT, each item of the input sequence is given a priority, and the model must output them in priority order.

We train each model in two regimes, one with a small number of samples (16K) and one with a large number of samples (320K). In the former case, the samples are iterated through 20 times, while in the latter, the samples are iterated through only once. Thus in both regimes, the total training times are the same. Training is done by minimizing negative log likelihood with RMSProp.

Prediction is performed via argmax/greedy prediction at each step. To evaluate the performance of the models, we compute the fraction of tokens correctly predicted and the fraction of all answers completely correctly predicted, respectively called fine and coarse scores. We assess the models on 3.2K randomly generated *out-of-sample* 2x length examples, i.e. with sequence lengths $2k$ (or repeat number $2N$ in the case of REPEAT COPY) to test the generalization of the system. More precisely, for all tasks other than repeat copy, during training, the length $k$ is varied in the interval $[l_k, u_k]$ (as shown in table 1ba). During test time, the length $k$ is varied in the range $[u_k + 1, 2u_k]$. For repeat copy, the repetition number $N$ is varied similarly, instead of $k$.

**Results.** Main results comparing the different memory systems and read computations on a series of tasks are shown in Table 1b. Consistent with previous work the fixed-memory LSTM system fails consistently when required to generalize to the 2x samples, unable to solve any 2x problem correctly, and only able to predict at most $\sim 50\%$ of the symbols for all tasks except interleaved addition, regardless of training regime. The RAM (attention-based) and the RAM/tape hybrid are much stronger baselines, answering more than $50\%$ of the characters correctly for all but the 6-ODD FIRST task. Perhaps surprisingly, RAM and RAM/tape learned the 7-REPEAT COPY task with almost perfect generalization scores when trained in the large sample regime. In general, it does not seem that the simple tape memory confers much advantage to the RAM model, as the generalization performances of both models are similar for the most part, which motivates more advanced NTM enhancements beyond sharpening.

The last four columns illustrate the performance of the LANTM models. We found the inverse-square LANTM and SLANTM models to be the most effective, achieving $> 90\%$ generalization

---

[6] Note that the read weight calculation of a SLANTM with softmax is essentially the same as the RAM model: For head $q$, $\exp(-d(q, k_i)^2/T) = \exp(-\|q - k_i\|^2/T) = \exp(-(2 - 2\langle q, k_i \rangle)/T)$, where the last equality comes from $\|q\| = \|k_i\| = 1$ (key-space is on the sphere). Therefore the weights $w_i = \frac{s_i \exp(-d(q,k_i)^2/T)}{\sum_j s_j \exp(-d(q,k_j)^2/T)} = \frac{s_i \exp(-2\langle q,k_i \rangle/T)}{\sum_j s_j \exp(-2\langle q,k_j \rangle/T)}$, which is the RAM weighting scheme.

| Task | Input | Output | Size $k$ | $|\mathcal{V}|$ |
|---|---|---|---|---|
| 1 - COPY | $a_1a_2a_3\cdots a_k$ | $a_1a_2a_3\cdots a_k$ | $[2,64]$ | 128 |
| 2 - REVERSE | $a_1a_2a_3\cdots a_k$ | $a_ka_{k-1}a_{k-2}\cdots a_1$ | $[2,64]$ | 128 |
| 3 - BIGRAM FLIP | $a_1a_2a_3a_4\cdots a_{2k-1}a_{2k}$ | $a_2a_1a_4a_3\cdots a_{2k}a_{2k-1}$ | $[1,16]$ | 128 |
| 4 - DOUBLE | $a_1a_2\cdots a_k$ | $2\times|a_k\cdots a_1|$ | $[2,40]$ | 10 |
| 5 - INTERLEAVED ADD | $a_1a_2a_3a_4\cdots a_{2k-1}a_{2k}$ | $|a_{2k}a_{2k-2}\cdots a_2|+|a_{2k-1}\cdots a_1|$ | $[2,16]$ | 10 |
| 6 - ODD FIRST | $a_1a_2a_3a_4\cdots a_{2k-1}a_{2k}$ | $a_1a_3\cdots a_{2k-1}a_2a_4\cdots a_{2k}$ | $[1,16]$ | 128 |
| 7 - REPEAT COPY | $\overline{N}a_1\cdots a_{20}$ | $a_1\cdots a_{20}\cdots a_1\cdots a_{20}$ ($N$ times) | $N\in[1,5]$ | 128 |
| 8 - PRIORITY SORT | $\overline{5}a_5\overline{2}a_2\overline{9}a_9\cdots$ | $a_1a_2a_3\cdots a_k$ | $[2,10]$ | 128 |

(a) Task descriptions and parameters. $|a_k\cdots a_1|$ means the decimal number repesented by decimal digits $a_k\cdots a_1$. Arithmetic tasks have all numbers formatted with the least significant digits *on the left* and with zero padding. The DOUBLE task takes an integer $x\in[0,10^k)$ padded to $k$ digits and outputs $2x$ in $k+1$ digits, zero padded to $k+1$ digits. The INTERLEAVED ADD task takes two integers $x,y\in[0,10^k)$ padded to $k$ digits and interleaved, forming a length $2k$ input sequence and outputs $x+y$ zero padded to $k+1$ digits. The last two tasks use numbers in unary format: $\overline{N}$ is the shorthand for a length $N$ sequence of a special symbol @, encoding $N$ in unary, e.g. $\overline{3}$ = @@@.

| | Base LSTM | | Memory RAM | | RAM/Tape | | LANTM | | LANTM-s | | Lie SLANTM | | SLANTM-s | |
|---|---|---|---|---|---|---|---|---|---|---|---|---|---|---|
| | S | L | S | L | S | L | S | L | S | L | S | L | S | L |
| 1 | 16/0 | 21/0 | 61/0 | 61/1 | 70/2 | 70/1 | ★ | ★ | ★ | ★ | ★ | ★ | ★ | ★ |
| 2 | 26/0 | 32/0 | 58/2 | 54/2 | 24/1 | 43/2 | ★ | ★ | 97/44 | 98/88 | 99/96 | ★ | ★ | ★ |
| 3 | 30/0 | 39/0 | 56/5 | 54/9 | 64/8 | 69/9 | ★ | ★ | ★ | 99/94 | 99/99 | 97/67 | 93/60 | 90/43 |
| 4 | 44/0 | 47/0 | 72/8 | 74/15 | 70/12 | 71/6 | ★ | ★ | ★ | ★ | ★ | ★ | ★ | ★ |
| 5 | 60/0 | 61/0 | 74/13 | 76/17 | 77/23 | 67/19 | **99/93** | 99/93 | 90/38 | 94/57 | 99/91 | 99/97 | 98/78 | ★ |
| 6 | 29/0 | 42/0 | 31/5 | 46/4 | 43/8 | 62/8 | **99/91** | **99/95** | 90/29 | 50/0 | 49/7 | 56/8 | 74/15 | 76/16 |
| 7 | 24/0 | 37/0 | 98/56 | **99/98** | 71/18 | 99/93 | 67/0 | 70/0 | 17/0 | 48/0 | **99/91** | 99/78 | 96/41 | 99/51 |
| 8 | 46/0 | 53/0 | 60/5 | 80/22 | 78/15 | 66/9 | 87/35 | 98/72 | 99/95 | 99/99 | ★ | 99/99 | 98/79 | ★ |

(b) Main results. Numbers represent the accuracy percentages on the fine/coarse evaluations on the out-of-sample 2× tasks. The S and L columns resp. indicate small and large sample training regimes. Symbol ★ indicates exact 100% accuracy (Fine scores above 99.5 are not rounded up). Baselines are described in the body. LANTM and SLANTM use inverse-square while LANTM-s and SLANTM-s use softmax weighting scheme. The best scores, if not 100% (denoted by stars), are bolded for each of the small and large sample regimes.

accuracy on most tasks, and together they solve all of the tasks here with $>90\%$ coarse score. In particular, LANTM is able to solve the 6-ODD FIRST problem when no other model can correctly solve 20% of the 2x instances; SLANTM on the other hand is the only Lie access model able to solve the 7-REPEAT COPY problem.

The best Lie access model trained with the small sample regime beats or is competitive with any of the baseline trained under the large sample regime. In all tasks other than 7-REPEAT COPY, the gap in the coarse score between the best Lie access model in small sample regime and the best baseline in any sample regime is $\geq 70\%$. However, in most cases, training under the large sample regime does not improve much. For a few tasks, small sample regime actually produces a model with better generalization than large sample regime. We observed in these instances, the generalization error curve under a large sample regime reaches an optimum at around 2/3 to 3/4 of training time, and then increases almost monotonically from there. Thus, the model likely has found an algorithm that works only for the training sizes; in particular, this phenomenon does not seem to be due to lack of training time.

## 6 DISCUSSION

**Qualitative Analysis.** We did further visual analysis of the different Lie-access techniques to see how the models were learning the underlying tasks, and to verify that they were using the relative addressing scheme. Figure 2 shows two diagrams of the LANTM model of the tasks of priority sort and repeat copy. Figure 3 shows two diagrams of the SLANTM model for the same two tasks. Fig-

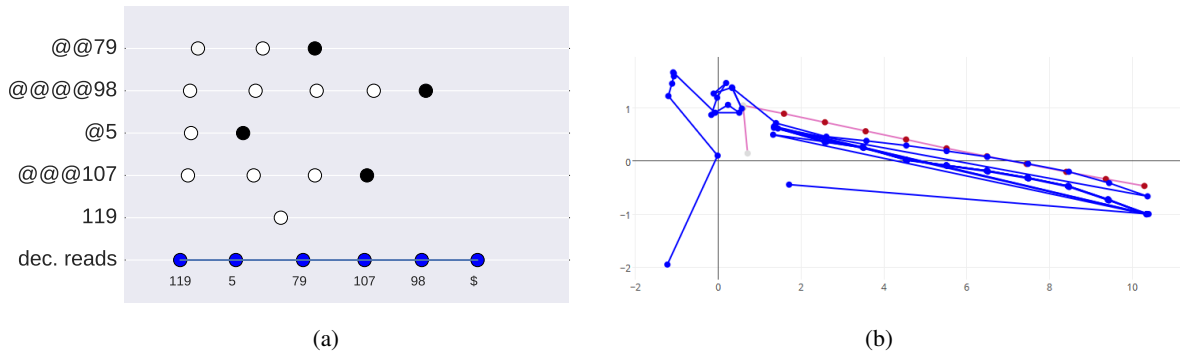

(a) (b)

Figure 2: Analysis of the LANTM model. **(a)** PCA projection from key space $\mathbb{R}^2$ to 1D for the memories $\Sigma$ and read heads $q$ of LANTM for the unary 8-PRIORITY SORT task. In this task, the encoder reads a priority, encoded in unary, and then a value; the decoder must output these values in priority order. In this example the sequence is $[@, @, 79, @, @, @, @, 98, @, 5, @, @, @, 107, @, 119]$, where the special symbol @ is a unary encoding of the priority. From top to bottom, each row indicates the movement of the encoder write head $q_{(w)}$ as it is fed each input character. Fill indicates the strength $s_i$ of memory write (black indicates high strength). Position of a dot within its row indicates the PCA projection of the key $k_i$. The last line indicates the movement of decoder read head $q$. Interestingly, we note that, instead of writing to memory, the controller remembers the item 119 itself. **(b)** Raw coordinates in key space $\mathbb{R}^2$ of writes (red) and reads (blue) from LANTM on 7-REPEAT COPY. Red line indicates the writes, which occur along a straight line during the encoding phase. Blue line indicates the reads, which zip back and forth in the process of copying the input sequence 6 times.

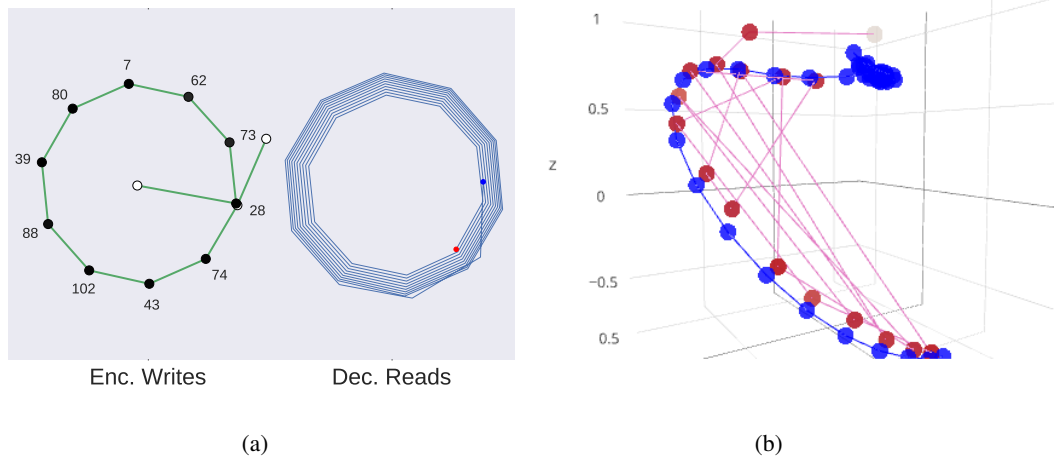

Enc. Writes          Dec. Reads

(a) (b)

Figure 3: Analysis of the SLANTM model. **(a)** PCA projection from the spherical key space $S^2$ to 2D of the memories $\Sigma$ and read heads $q$ of SLANTM for the task of 7-REPEAT COPY. Here the model is to repeatedly output the sequence 10 times. Input is 10 repetitions of special symbol @ followed by [28, 74, 43, 102, 88, 39, ... ]. *Left*: the positions of write head $q_{(w)}$ during the encoding phase. Fill indicates strength $s_i$ (black means high strength); number indicates the character stored. SLANTM traverses in a circle clockwise starting at point 28, and stores data at regular intervals. *Right*: the positions of read head $q$ during the decoding phase. Starting from the blue dot, the reads move clockwise around the sphere, and end at the red dot. For the sake of clarity, read positions are indicated by bends in the blue line, instead of by dots. Intriguingly, the model implements a cyclic list data structure, taking advantage of the spherical structure of the memory. **(b)** Raw coordinates in key space $S^2$ of writes (red) and reads (blue) from SLANTM on a non-unary encoded variant of the priority sort task. Red line indicates the movements of the write-head $q_{(w)}$ to place points along a sub-manifold of $\mathcal{K}$ (an arc of $S^2$) during the encoding phase. Notably, this movement is not sequential, but random-access, so as to store elements in correct priority order. Blue line indicates the simple traversal of this arc during decoding.

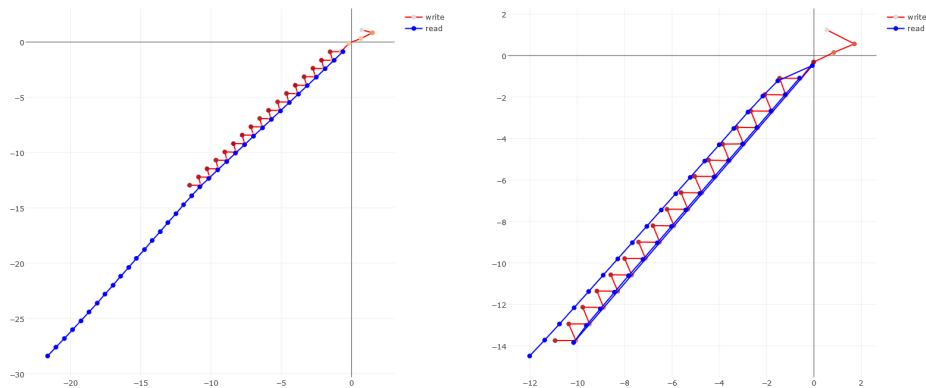

Figure 4: Memory access pattern of LANTM on 6-Odd First. Left: In the middle of training. LANTM learns to store data in a zigzag such that odd-indexed items fall on one side and even-indexed items fall on the other. However reading is only half correct. Right: After training. During reading, the model simply reads the odd-indexed items in a straight line, followed by the even-indexed items in a parallel line.

ure 4 shows the memory access pattern of LANTM on 6-Odd First task. Additionally, animations tracing the evolution of the memory access pattern of models over training time can be found at http://nlp.seas.harvard.edu/lantm. They demonstrate that the models indeed learn relative addressing and internally are constructing geometric data structures to solve these algorithmic tasks.

**Unbounded storage**    One possible criticism of the LANTM framework could be that the amount of information stored increases linearly with time, which limits the usefulness of this framework for long timescale tasks. This is indeed the case with our implementations, but need not be the case in general. There can be many ways of limiting physical memory usage. For example, a simple way is to discard the least recently used memory, as in the work of Graves et al. (2016) and Gulcehre et al. (2016). Another way is to approximate with fixed number of bits the read function that takes a head position and returns the read value. For example, noting that this function is a rational function on the head position, keys, and memory vectors, we can approximate the numerators and denominators with a fixed degree polynomial.

**Content address**    Our Lie-access framework is not mutually exclusive from content addressing methods. For example, in each of our implementations, we could have the controllers output both a position in the key space and a content addresser of the same size as memory vectors, and interpolated the read values from Lie-access and the read values from content addressing.

## 7    CONCLUSION

This paper introduces Lie-access memory as an alternative neural memory access paradigm, and explored several different implementations of this approach. LANTMs follow similar axioms as discrete Turing machines while providing differentiability. Experiments show that simple models can learn algorithmic tasks. Internally these models naturally learn equivalence of standard data structures like stack and cyclic lists. In future work we hope to experiment with more groups and to scale these methods to more difficult reasoning tasks. For instance, we hope to build a general purpose encoder-decoder model for tasks like question answering and machine translation that makes use of differentiable relative-addressing schemes to replace RAM-style attention.

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

# Appendices

## A    EXPERIMENTAL DETAILS

We obtain our results by performing a grid search over the hyperparameters specified in Table A.1 and also over seeds 1 to 3, and take the best scores. We bound the norm of the LANTM head shifts by 1, whereas we try both bounding and not bounding the angle of rotation in our grid for SLANTM. We initialize the Lie access models to favor Lie access over random access through the interpolation mechanism discussed in section 4.3.

The RAM model read mechanism is as discussed in section 2, and writing is done by appending new $(k, v, s)$ tuples to the memory $\Sigma$. The only additions to this model in RAM/tape is that left and right keys are now computed using shifted convolution with the read weights:

$$k_L := \sum_i w_{i+1} k_i$$

$$k_R := \sum_i w_{i-1} k_i$$

and these keys $k_L$ and $k_R$ are available (along with the random access key output by the controller) to the controller on the next turn to select from via interpolation. We also considered weight sharpening in the RAM/Tape model according to Graves et al. (2014): the controller can output a *sharpening coefficient* $\gamma \geq 1$ each turn, so that the final weights are $\tilde{w}_i = \frac{w_i^\gamma}{\sum_j w_j^\gamma}$. We included this as a feature to grid search over.

|  | rnn size | embed | decay delay | init | learning rate | key dim | custom |
|---|---|---|---|---|---|---|---|
| LANTM(-s) | $50 \times 1$ | 14 | $\{300, 600\}$ | $\{1, *\}$ | $\{1, 2, 4\}$e-2 | 2 | - |
| SLANTM(-s) | $50 \times 1$ | 14 | $\{300, 600\}$ | $\{1, *\}$ | $\{1, 2, 4\}$e-2 | 3 | $\angle$ bound |
| RAM(/tape) | $50 \times 1$ | 14 | $\{300, 600\}$ | $\{1, *\}$ | $\{1, 2, 4\}$e-2 | $\{2, 20\}$ | sharpen |
| LSTM | $256 \times \{1 \text{ to } 4\}$ | 128 | $\{500, 700\}$ | $*$ | 2e-$\{1 \text{ to } 4\}$ | - | - |

Table A.1: Parameter grid for grid search. LANTM(-s) means LANTM with invnorm or SoftMax; similarly for SLANTM(-s). RAM(/tape) means the ram and hybrid ram/tape models. Initialization: both initialization options set the forget gate of the LSTMs to 1. The number 1 in the init column means initialization of all other parameters uniformly from $[-1, 1]$. The symbol * in init column means initialization of all linear layers were done using the torch default, which initializes weights uniformly from $(-\kappa, \kappa)$, where $\kappa$ is (input size)$^{-1/2}$. For models with memory, this means that the LSTM input to hidden layer is initialized approximately from $[-0.07, 0.07]$ (other than forget gate). Angle bound is a setting only available in SLANTM. If angle bound is true, we bound the angle of rotation by a learnable magnitude value. Sharpening is a setting only available in RAM/tape, and it works as explained in the main text.

We found that weight sharpening only confers small advantage over vanilla on the COPY, BIGRAM FLIP, and DOUBLE tasks, but deteriorates performance on all other tasks.

## B    ACTION INTERPOLATION

We also experimented with adding an interpolation between the last action $a^{(t-1)}$ with a candidate action $a(h)$ via a gate $r(h) \in [0, 1]$ to produce the final action $a^{(t)}$. Then the final equation of the new head is

$$q' := a^{(t)} \cdot \pi(tq + (1 - t)\tilde{q}).$$

This allows the controller to easily move in "a straight line" by just saturating both $t$ and $r$.

For example, for the translation group we have straight-line interpolation, $a^{(t)} := ra + (1-r)a^{(t-1)}$. For the rotation group $SO(3)$, each rotation is represented by its axis $\xi \in S^2$ and angle $\theta \in (-\pi, \pi]$,

|   | LANTM | | LANTM-s | | SLANTM | | SLANTM-s | |
|---|---|---|---|---|---|---|---|---|
|   | S | L | S | L | S | L | S | L |
| 1 | ⋆:⋆/⋆:⋆ | ⋆:⋆/⋆:⋆ | ⋆:⋆/⋆:⋆ | ⋆:⋆/⋆:⋆ | ⋆:⋆/⋆:⋆ | ⋆:⋆/⋆:⋆ | ⋆:99/⋆:83 | ⋆:99/⋆:99 |
| 2 | ⋆:⋆/⋆:⋆ | ⋆:⋆/⋆:⋆ | 97:85/44:60 | 98:91/88:55 | 99:99/96:98 | ⋆:⋆/⋆:⋆ | ⋆:⋆/⋆:⋆ | ⋆:⋆/⋆:⋆ |
| 3 | ⋆:⋆/⋆:⋆ | ⋆:99/⋆:77 | ⋆:99/⋆:93 | 99:92/94:17 | 99:⋆/99:⋆ | 97:99/67:73 | 93:99/60:62 | 90:92/43:57 |
| 4 | ⋆:⋆/⋆:⋆ | ⋆:⋆/⋆:⋆ | ⋆:⋆/⋆:⋆ | ⋆:⋆/⋆:⋆ | ⋆:⋆/⋆:⋆ | ⋆:⋆/⋆:⋆ | ⋆:⋆/⋆:⋆ | ⋆:⋆/⋆:⋆ |
| 5 | 99:⋆/93:⋆ | 99:⋆/93:⋆ | 90:99/38:80 | 94:99/57:84 | 99:96/91:61 | 99:99/97:99 | 98:99/78:99 | ⋆:⋆/⋆:⋆ |
| 6 | 99:50/91:0 | 99:54/95:0 | 90:56/29:0 | 50:57/0:0 | 49:73/7:33 | 56:76/8:27 | 74:92/15:45 | 76:81/16:31 |
| 7 | 67:52/0:0 | 70:22/0:0 | 17:82/0:0 | 48:98/0:8 | 99:⋆/91:⋆ | 99:97/78:21 | 96:90/41:22 | 99:99/51:99 |
| 8 | 87:97/35:76 | 98:93/72:38 | 99:81/95:24 | 99:50/99:0 | ⋆:99/⋆:99 | 99:99/99:95 | 98:95/79:60 | ⋆:98/⋆:80 |

Table B.2: Comparison between scores of model with action interpolation and without action interpolation. Numbers represent the accuracy percentages on the fine/coarse evaluations on the out-of-sample $2\times$ tasks. The S and L columns resp. indicate small and large sample training regimes. Symbol ⋆ indicates exact 100% accuracy (Fine scores above 99.5 are not rounded up). Each entry is of the format A:B/C:D, where A and C are respectively the fine and coarse scores of the model without action interpolation (same as in table 1b), and B and C are those for the model with action interpolation.

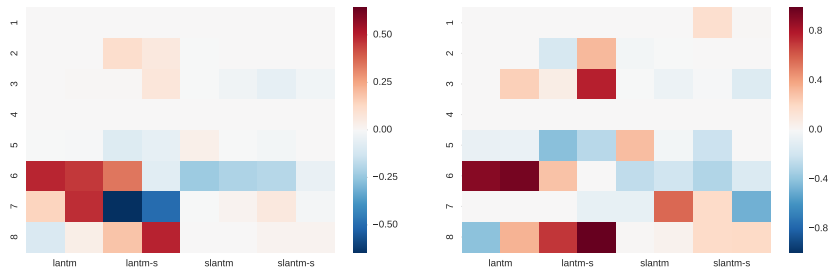

Figure B.1: The additive difference between the fine (left) and coarse (right) scores of models without action interpolation vs models with action interpolation. Positive value means model without interpolation performs better. For each model, the left column displays the difference in small sample regime, while the right column displays the difference in large sample regime.

and we just interpolate each separately $\xi^{(t)} := \pi(r\xi + (1-r)\xi^{(t-1)})$ and $\theta^{(t)} := r\theta + (1-r)\theta^{(t-1)}$. where $\pi$ is $L_2$-normalization.[7]

We perform the same experiments, with the same grid as specified in the last section, and with the initial action interpolation gates biased toward the previous action. The results are given in table B.2. Figure B.1 shows action interpolation's impact on performance. Most notably, interpolation seems to improve performance of most models in the 5-INTERLEAVED ADD task and of the spherical memory models in the 6-ODD FIRST task, but causes failure to learn in many situations, most significantly, the failure of LANTM to learn 6-ODD FIRST.

---

[7]There is, in fact, a canonical way to interpolate the most common Lie groups, including all of the groups mentioned above, based on the exponential map and the Baker-Campbell-Hausdorff formula (Lee, 2012), but the details are outside the scope of this paper and the computational cost, while acceptable in control theory settings, is too hefty for us. Interested readers are referred to Shingel (2009) and Marthinsen (1999).

