# Peer review of "Lie-Access Neural Turing Machines"

_ICLR 2017 — accepted_

[Public Comment · Tara N Sainath · 07 Nov 2016]
**ICLR Paper Format**

Dear Authors,

Please resubmit your paper in the ICLR 2017 format with the correct font for your submission to be considered. Thank you!

[Reviewer Comment · AnonReviewer2 · 01 Dec 2016]
**Figure 2 difficult to make sense of**
soundness 2 · originality 2 · appropriateness 2

I struggle to understand figure 2, despite the length of the caption. Perhaps labelling the images themselves a bit more clearly.

[Official Review · AnonReviewer4 · rating 8 · confidence 4 · 15 Dec 2016]
**This paper brings unity and formalism in the requirement for memory addressing while maintaining differentiable memories. Its proposal provide a generic scheme to build addressing mechanisms. When comparing the proposed approach with key-value networks, the unbounded number of memory cells and the lack of incentive to reuse indexes might reveal impractical.**

*** Paper Summary ***

This paper formalizes the properties required for addressing (indexing) memory augmented neural networks as well as how to pair the addressing with read/write operation. It then proposes a framework in which any Lie group as the addressing space. Experiments on algorithmic tasks are reported.

*** Review Summary ***

This paper brings unity and formalism in the requirement for memory addressing while maintaining differentiable memories. Its proposal provide a generic scheme to build addressing mechanisms. When comparing the proposed approach with key-value networks, the unbounded number of memory cells and the lack of incentive to reuse indexes might reveal impractical. 

*** Detailed Review ***

The paper reads well, has appropriate relevance to related work. The unified presentation of memory augmented networks is clear and brings unity to the field. The proposed approach is introduced clearly, is powerful and gives a tool that can be reused after reading the article. I do not appreciate that the growing memory is not mentioned as a drawback. It should be stressed and a discussion on the impact it has on efficiency/scalability is needed.

[Official Review · AnonReviewer3 · rating 6 · confidence 3 · 16 Dec 2016]
**interesting new**
soundness 3 · originality 2 · impact 5

The paper proposes a new memory access scheme based on Lie group actions for NTMs.

Pros:
* Well written
* Novel addressing scheme as an extension to NTM.
* Seems to work slightly better than normal NTMs.
* Some interesting theory about the novel addressing scheme based on Lie groups.

Cons:
* In the results, the LANTM only seems to be slightly better than the normal NTM.
* The result tables are a bit confusing.
* No source code available.
* The difference to the properties of normal NTM doesn't become too clear. Esp it is said that LANTM are better than NTM because they are differentiable end-to-end and provide a robust relative indexing scheme but NTM are also differentiable end-to-end and also provide a robust indexing scheme.
* It is said that the head is discrete in NTM but actually it is in space R^n, i.e. it is already continuous. It doesn't become clear what is meant here.
* No tests on real-world tasks, only some toy tasks.
* No comparisons to some of the other NTM extensions such as D-NTM or Sparse Access Memory (SAM) (

[Official Review · AnonReviewer5 · rating 6 · confidence 4 · 17 Dec 2016]

The paper introduces a novel memory mechanism for NTMs based on differentiable Lie groups. 
This allows to place memory elements as points on a manifold, while still allowing training with backpropagation.
It's a more general version of the NTM memory, and possibly allows for training a more efficient addressing schemes.

Pros:
- novel and interesting idea for memory access
- nicely written
 
Cons:
- need to manually specify the Lie group to use (it would be better if network could learn the best way of accessing memory)                                 
- not clear if this really works better than standard NTM (compared only to simplified version)
- not clear if this is useful in practice (no comparison on real tasks)

[Official Review · AnonReviewer2 · rating 7 · confidence 4 · 17 Dec 2016]
**mathematically elegant, limited impact**
soundness 2 · originality 2 · appropriateness 2

The Neural Turing Machine and related “external memory models” have demonstrated an ability to learn algorithmic solutions by utilizing differentiable analogues of conventional memory structures. In particular, the NTM, DNC and other approaches provide mechanisms for shifting a memory access head to linked memories from the current read position.

The NTM, which is the most relevant to this work, uses a differentiable version of a Turing machine tape. The controller outputs a kernel which “softly” shifts the head, allowing the machine to read and write sequences. Since this soft shift typically “smears” the focus of the head, the controller also outputs a sharpening parameter which compensates by refocusing the distribution.

The premise of this work is to notice that while the NTM emulates a differentiable version of a Turing tape, there is no particular reason that one is constrained to follow the topology of a Turing tape. Instead they propose memory stored at a set of points on a manifold and shift actions which form a Lie group. In this way, memory points can have have different relationships to one another, rather than being constrained to Z.

This is mathematically elegant and here they empirically test models with the shift group R^2 acting on R^2 and the rotation group acting on a sphere.

Overall, the paper is well communicated and a novel idea.

The primary limitation of this paper is its limited impact. While this approach is certainly mathematically elegant, even likely beneficial for some specific problems where the problem structure matches the group structure, it is not clear that this significantly contributes to building models capable of more general program learning. Instead, it is likely to make an already complex and slow model such as the NTM even slower. In general, it would seem memory topology is problem specific and should therefore be learned rather than specified.

The baseline used for comparison is a very simple model, which does not even having the sharpening (the NTM approach to solving the problem of head distributions becoming ‘smeared’). There is also no comparison with the successor to the NTM, the DNC, which provides a more general approach to linking memories based on prior memory accesses.

Minor issues:
Footnote on page 3 is misleading regarding the DNC. While the linkage matrix explicitly excludes the identity, the controller can keep the head in the same position by gating the following of the link matrix.
Figures on page 8 are difficult to follow.

[Author Response · Greg Yang · 13 Jan 2017]
**Sharpening results.**

We have finished results adding sharpening to the RAM/tape model. Weight sharpening only confers small advantage over vanilla on the copy, bigram flip, and double tasks, but deteriorates performance on all other tasks. These results will appear in the final version of the paper.

[Final Decision · Program Chairs · 06 Feb 2017]
**ICLR committee final decision**

The paper presents a Lie-(group) access neural turing machine (LANTM) architecture, and demonstrates it's utility on several problems. 
 
 Pros:
 Reviewers agree that this is an interesting and clearly-presented idea.
 Overall, the paper is clearly written and presents original ideas.
 It is likely to inspire further work into more effective generalizations of NTMs.
 
 Cons:
 The true impact and capabilities of these architectures are not yet clear, although it is argued that the same can be said for NTMs. 
 
 The paper has been revised to address some NTM features (sharpening) that were not included in the original version.
 The purpose and precise definition of the invNorm have also been fixed.